# Effect of Polyploidy Induction on Natural Metabolite Production in Medicinal Plants

**DOI:** 10.3390/biom11060899

**Published:** 2021-06-17

**Authors:** Hadi Madani, Ainoa Escrich, Bahman Hosseini, Raul Sanchez-Muñoz, Abbas Khojasteh, Javier Palazon

**Affiliations:** 1Department of Horticulture, Faculty of Agriculture, Urmia University, Urmia 5756151818, Iran; Madani75@gmail.com (H.M.); b.hosseini@urmia.ac.ir (B.H.); 2Department of Experimental and Health Sciences, Universitat Pompeu Fabra, 08003 Barcelona, Spain; ainoa.escrich@upf.edu; 3Laboratory of Functional Plant Biology, Department of Biology, Ghent University, K.L. Ledeganckststraat 35, B-9000 Ghent, Belgium; raul.sanchezmunoz@ugent.be; 4Department of Plant Physiology, Faculty of Pharmacy, University of Barcelona, Av. Joan, XXIII, 08028 Barcelona, Spain; abbaskhojasteh@aol.com

**Keywords:** colchicine, gene expression, medicinal plants, polyploidy, specialized plant metabolites

## Abstract

Polyploidy plays an important role in plant diversification and speciation. The ploidy level of plants is associated with morphological and biochemical characteristics, and its modification has been used as a strategy to alter the quantitative and qualitative patterns of secondary metabolite production in different medicinal plants. Polyploidization can be induced by many anti-mitotic agents, among which colchicine, oryzalin, and trifluralin are the most common. Other variables involved in the induction process include the culture media, explant types, and exposure times. Due to the effects of polyploidization on plant growth and development, chromosome doubling has been applied in plant breeding to increase the levels of target compounds and improve morphological characteristics. Prompted by the importance of herbal medicines and the increasing demand for drugs based on plant secondary metabolites, this review presents an overview of how polyploidy can be used to enhance metabolite production in medicinal plants.

## 1. Importance of Medicinal Plants

As an essential part of our daily diet, plants play a central role in human health. In addition to their nutritional value, plants have been intensively studied for their bioactive compounds [1]. In addition to essential primary metabolites, plants are able to synthesize a wide variety of compounds, known as secondary metabolites or specialized plant metabolites, which have important roles in plant interactions with the environment [2]. They are usually produced in low amounts (less than 1% dry weight), and their production varies depending on the physiological and developmental stages of the plant, as well as environmental factors. In addition, many secondary metabolites are restricted to only a few species, and even among these, the biosynthetic machinery and its specific regulation can be highly diverse [3]. Therefore, the study of specialized plant metabolic pathways and their regulation and the optimization of metabolite production are open research fields.

Notable examples of secondary metabolites used as medicines are morphine, codeine, cocaine, tropane alkaloids (hyoscyamine and scopolamine), colchicine, physostigmine, pilocarpine, reserpine, berberine, paclitaxel, tubocurarine, and quinine, as well as anti-cancer alkaloids derived from *Catharanthus roseus* (vinblastine and vincristine) and steroids such as diosgenin, digoxin, and digitoxin [4]. Although new methods for drug discovery and development have emerged, such as chemo-typing and computer models for molecular design, natural compounds continue to play a predominant role. Currently, one-fourth of all prescribed pharmaceuticals in industrialized countries contain compounds that are directly or indirectly (via semi-synthesis) isolated from plants. The World Health Organization has estimated that up to 80% of people still rely on herbal remedies for their health care [5], and plant-derived drugs have a huge market value. The output of Chinese Materia Medica was estimated to be worth USD 83.1 billion in 2012, an increase of more than 20% from the previous year [5]. In the Republic of Korea, annual spending on traditional medicine was USD 4.4 billion in 2004, reaching USD 7.4 billion in 2009 [5]. Expenditure on natural products in the United States was USD 14.8 billion in 2008 [6]. It is estimated that the global trade in medicinal plants will be worth over USD 5 trillion by 2050. In addition to the central importance of natural products as sources of pharmaceutical agents and their direct use as drugs, they can also be used as chemical models for the design and synthesis of new drug substances.

Despite their significance, only a limited number of medicinal plants are cultivated, most of which are collected from the wild. However, overharvesting plants in their natural habitat can contribute to a loss of genetic diversity and habitat destruction. As an alternative, agricultural cultivation of medicinal plants offers several advantages: reliable botanical identification; greater genetic, phenotypic, and phytochemical control; production of cultivars modified according to market requirements; less pressure on wild plants; and a more stable source of raw material. Nevertheless, the domestication of medicinal plants and the development of agronomic cultures are not always possible and can lead to a dramatic modification of the content of bioactive compounds compared to wild-type plants [7].

In agronomic research and development, many biotechnological strategies have been tested to enhance the production of secondary metabolites in medicinal plants, including screening for high-yielding cell lines, modification of culture media, precursor feeding, elicitation, hairy root culture, plant cell immobilization, and biotransformation [8,9]. Cell cultures have been established from many plant species, but they often fail to produce sufficient amounts of the target compounds [10,11]. Strategies to improve yields include the treatment of undifferentiated cells with elicitors such as methyl jasmonate, salicylic acid, chitosan, and heavy metals, among others [12,13]. In some cases, secondary metabolites are only produced in organ cultures; for example, hairy roots produce high levels of alkaloids [14], whereas shooty teratoma (tumor-like) cultures produce monoterpenes [15].

Plant biofactories are biotechnological platforms based on plant cell and organ cultures that can be used for the production of pharmaceutical and biopharmaceutical compounds. However, to date, few plant cell factories have been successfully established to produce high-value secondary metabolites at industrial levels. These include cell suspension cultures of *Lithospermum erythrorhizon*, which is used for the production of shikonin, and *Coptis japonica*, which produces berberine [16]. Large-scale rosmarinic acid production has been achieved with cell cultures of *Coleus blumeii*, and sanguinarine, which has market potential in oral hygiene products, has been obtained in cell cultures of *Papaver somniferum* [17,18]. An example of a valuable pharmaceutical product successfully obtained from plant cell cultures is paclitaxel, an anti-cancer agent originally extracted from the bark of 50–60-year-old Pacific yew trees (*Taxus brevifolia*). Effective secondary metabolite production in cell or organ cultures requires several technical bottlenecks to be overcome, including low productivity and technological issues (e.g., bioreactor design and culture conditions). 

The breeding of cultivars enables genotypes to be adapted to market requirements. Breeding strategies can increase yields of valuable metabolites, eliminate unwanted compounds, enhance tolerance to abiotic and biotic stresses, and achieve homogeneity. Compared with traditional food crops, the breeding of medicinal plants is still at an early stage. However, the high natural genetic variability within one species is an advantage for breeders, as it can be exploited to obtain a selection response in a relatively short time [19]. Therefore, preventing the loss of genetic diversity is important. Natural variability within a species can be harnessed by gathering accessions from the wild, germplasm collections, or cultivars [20]. Examples of important medicinal plants that have been subjected to breeding programs are *Artemisia annua*, *Artemisia umbelliformis*, *Hypericum perforatum*, and *Thymus vulgaris*. 

Polyploidy, a type of mutation that increases the number of chromosomes, is a widespread phenomenon in the evolution, development, agriculture, and diversity of flowering plants [19]. The possession of multiple sets of chromosomes as a consequence of whole-genome duplication has been studied for over 100 years. Higher ploidy organisms often exhibit morphological features that differ from or are more pronounced than those of their diploid progenies [21]. Duplicated genes can be lost, retained, or maintained as duplicates, often undergoing subfunctionalization and neofunctionalization, which leads to morphological and functional changes [22,23]. The organization and function of the genome at both genetic and epigenetic levels are influenced by polyploidy. Epigenetic control can modulate the activation or suppression of gene expression. 

The advantages of polyploidy are gene redundancy and heterosis, which are the result of gene duplication. The principal disadvantages include the disrupting effects of nuclear and cell enlargement, the epigenetic instability that results in the alteration of gene regulation, and errors in meiosis. Thus, polyploidy may affect specialized plant metabolites [21], resulting in an increase or decrease in their production depending on the plant species. This review discusses polyploidy as an approach to enhancing secondary metabolite production, with a focus on induction methods and their application in medicinal plants.

## 2. The Importance of Polyploidy

Polyploidy is an important tool in plant breeding, as it results in increased genetic diversity. The traits of a plant can be altered by changing the chromosomal groups and the number of genes in a cell, with an outcome that may be desirable or undesirable. Polyploidy occurs in more than 80% of plant species and is responsible for 2–4% of speciation in flowering plants. Domesticated crops such as durum wheat, cotton, tobacco, and potatoes are polyploid organisms, as are ornamental flowers such as violets and lily of the valley [24]. Polyploids can be categorized as auto- or allopolyploids according to their origin, as described below.

### 2.1. Autopolyploids 

Autopolyploids are established when the cell cannot divide properly, and the chromosomes are doubled but not separated, so all of their chromosomes are derived from the same species. Spontaneous non-disjunction of chromosomes (Figure 1) caused by meiotic failure triggers a whole-genome duplication event, in which a diploid (2*n* = 2x) is transformed into an autotetraploid (2*n* = 4x) organism. Autopolyploidy can also be induced through environmental factors, chemicals, and laboratory techniques [25,26]. In addition to occurring naturally in many genera, autopolyploidy has been induced in plants to promote desired characteristics (Figure 1B,C) [24,25,26,27,28]. Polyploidy has been used by breeders to improve crops. Tetraploid variants may present beneficial characteristics, such as larger fruit size, the absence of seeds, high productivity, and conspicuous changes in plant secondary metabolism [29,30]. For example, autotetraploid watermelons have been bred to obtain seedless fruits [31,32]. Polyploidy can be achieved through spontaneous natural mechanisms, such as meiotic or mitotic failure and fusion of unreduced (2*n*) gametes [21], or by treating diploids with mitotic inhibitors such as dinitroanilines and colchicine [33].

### 2.2. Allopolyploids 

Allopolyploids have more than two chromosome sets derived from different plant species and are generated by hybrid polyploidization (Figure 2). Polyploidy, a natural force in plant evolution, has been applied with hybridization to obtain populations with greater adaptive ability and growth potential. Hybrids have a set of chromosomes from each parent, but as these are not fully homologous and/or have unbalanced chromosome sets, meiosis is impeded, and most hybrids are sterile [21]. Thus, after artificially doubling the chromosomes so that somatic cells have two chromosome sets from each parent, meiosis can occur naturally, and viable gametes are generated. The resulting allopolyploid cannot reproduce with the parents and becomes a new species [21,22]. Accordingly, allopolyploids have been successfully used to introduce beneficial foreign genes belonging to two or more different plant species. Most plant hybrids are considered allopolyploids, including cultured hybrids artificially chosen for their larger cells, plump plant parts, high water content, and drought resistance [33]. Despite several problems encountered in interspecific gene transfer, artificial allopolyploidy has been effectively applied to restore fertility, overcome heteroploid cross-incompatibility by ensuring compatible ploidy levels in the prospective parents, and regulate chromosome pairing to incorporate target chromosomes [34].

## 3. The Importance of Polyploidy in Medicinal Plant Breeding

The exploitation of the genetic material of medicinal plants is still in its early stages. In polyploid plants, the sizes of flowers, leaves, fruits, and seeds are often increased [35]. The enzyme activity and the production of secondary metabolites responsible for taste or bioactive properties can also be enhanced by polyploidy [36]. The multiplication of the genome and the number of genes in autopolyploids can result in a higher metabolite yield [29]. For plant-breeding purposes and genetic studies, a set of chromosomes can be used to modify or produce new genotypes of cultivars [37]. Polyploidy plants cannot always be segregated phenotypically from their diploid parents and could have a different phenotype; therefore, they are not restricted by the traits of ancestral diploids and may have different levels of resistance to drought and insects, biomass production, and quality and concentration of bioactive plant compounds [38].

## 4. Methods of Increasing Ploidy Levels in Plants

Polyploid organisms can arise in two ways: spontaneous formation and artificial induction by chemical and physical agents. Most chemical mutagens are alkylating agents and azides. Physical mutagens include electromagnetic radiation, such as X-rays, UV light, and radiation of particles such as fast and thermal neutrons and β and α particles [39]. However, the efficiency and success rate of polyploidy induction in plants is low due to factors such as undesirable chromosomal changes, the creation of chimeric plants, and a lack of root production or even the death of treated plants. This section explores different techniques to increase the chromosomal level in plants. 

The synthetic duplication of chromosomes in plants can be achieved by interfering with the plant cell cycle (Figure 3), which can be divided into different phases: the G1-phase (post-mitotic interphase), the S-phase (DNA synthesis phase), the G2-phase (pre-mitotic interphase), and the M-phase (mitosis) [40]. Hence, between the S-phase and the end of mitosis, the cell duplicates its complement of chromosomes. During metaphase, the cylindrical microtubules become attached to proteins in the kinetochore and pull the chromosomes to the metaphase plate; in the subsequent anaphase, they move the sister chromatids to each pole of the cell [41]. The consequence of this chromosomal separation is the doubling of cells with the same set of chromosomes. In contrast, anti-mitotic agents, such as colchicine, oryzalin, nitroxide, lactacystin, MG132, and epoxomicin, generate cells with different chromosome sets [37].

Mutagenic agents such as colchicine, oryzalin, nitroxide, and epoxomicin have been used for polyploidy induction in plants. The most frequently applied is colchicine, although it sometimes has undesirable side effects, including sterility, abnormal growth, chromosomal rearrangement or reduction, and gene mutation [37,42,43,44,45]. Various explants, such as lateral buds, leaflets, nodules, small branches, tuberous and rhizome fragments, seedlings, sexual or somatic embryos, calli, and cell suspension cultures, have been treated with anti-mitotic agents to generate polyploid plants. During in vitro induction, the culture media can be solid, semi-solid, or liquid, depending on the plant species and anti-mitotic agent. The duration of the treatment also depends on the anti-mitotic agent used, the plant species, the culture medium, and the explant size and type [37,38,39,40,41,42,43,44,45].

## 5. Polyploidy Effects on Gene Expression and Silencing

Polyploidy has significant effects on the co-expression of duplicated genes, potentially eliminating, reducing, or increasing gene expression. These changes can occur with the onset of polyploidy or after several generations and can also be influenced by epigenetic factors. On the other hand, many changes in gene expression may occur only in a specific organ, so the relative expression of duplicated genes can vary in different parts of the plant. 

In polyploids, the activity pattern of genes varies greatly. Studies on *Arabidopsis* plants have shown that some of the genes that were active in the diploid state were shut down in the autotetraploid state, but the cross between two tetraploid *Arabidopsis* species led to the production of allotetraploid plants in which these genes reactivated [46]. In addition, a gene called rad54, which was not expressed in diploid *Arabidopsis* leaves, was activated in the leaves in the autotetraploid state [47]. Recent experiments demonstrated that the various ploidy levels in a plant (autoploids or alloploids) and even the previous generations play a role in the level and stability of gene expression [23]. There is considerable discussion about the factors that affect the activity of genes and their expression in polyploidy, including the activation of dormant transpositions in synthetic polyploids, which can cause gene extinction [48]. Other factors that silence duplicated genes in polyploids include deacetylation, methylation, and changes in histones and chromatin structure [49].

Small RNAs such as miRNAs (microRNAs), siRNAs (small interfering RNAs), and ta-siRNAs (trans-acting siRNAs) appear to play an important role in regulating gene expression and epigenetic changes induced by polyploidy [50]. Some studies report a high overall gene expression in plants with increased ploidy levels, including *Atropa belladonna* [51], *Papaver somniferum* [52], and *Artemisia annua* [53]. The expression levels of some key genes involved in the podophyllotoxin biosynthetic pathway, including those encoding phenylalanine ammonia-lyase (PAL), cinnamoyl-CoA reductase (CCR), cinnamyl-alcohol dehydrogenase (CAD), and pinoresinol-lariciresinol reductase (PLR), were enhanced by increasing the plant ploidy [54]. In research on *A. annua* L., the expression of genes encoding 10 key enzymes involved in the artemisinin biosynthetic pathway was analyzed in plants of different ploidy levels [53]. The results show that, in tetraploid plants, the expression of some key genes in the artemisinin biosynthetic pathway may be positively regulated [53]. Similarly, analysis through semi-quantitative reverse transcription–polymerase chain reaction (RT-PCR) of various genes involved in morphinane biosynthesis showed increased expression in tetraploid plants of *Papaver somniferum* [52]. Based on the above discussion, it is concluded that polyploidy activates cell mechanisms, affecting the amount of DNA template and its translation and transcription and resulting in increased, decreased, or even silenced gene expression. 

## 6. Polyploidy Effects on Valuable Compounds

Artificial polyploidy generally enhances the vigor of specific plant parts and may be favorable when specific organs and/or biomass constitute the commercial product [55]. Polyploidy has been used to increase plant secondary metabolite production or to improve the metabolite profile. Evidence from the biochemical analysis of many allopolyploid plants suggests that their enzymatic capacity is higher compared to the parental individuals, and they are also richer in phenolic compounds [55]. Since autopolyploids are characterized by direct genomic duplication, the genotype remains constant, yet the genetic material is multiplied. This can lead to increased activity of previously weakly expressed genes in a target biosynthetic pathway and improve metabolite production. 

The induction of polyploidy can influence the physiological and biochemical processes of the plant and affect the biosynthetic pathways of primary and secondary metabolites. A common explanation for these changes is the reduction in the ratio of the membrane to the amount of chromatin, which leads to increased contact between the chromatin material and the nuclear membrane and enhanced activity of the gene for each cell. The volume and level of tetraploid versus diploid cell proliferation are 2 and 1.5, respectively, so it is an advantage if cellular production is related to the metabolic activity of the cell surface.

Increased ploidy levels are also associated with gene alteration and rearrangement. Changing the expression of genes involved in the biosynthetic pathways of secondary metabolites may alter the constraints on the expression of a key gene. Moreover, turning the gene on or off can lead to increased production of compounds or specific formulations, as well as changes in the pattern of production. Several examples are given in Table 1. Interestingly, an altered chemical profile of the metabolic products is frequently encountered. Polyploidy influences the evolution of both the primary and secondary metabolism of plants (Table 1) [56].

### 6.1. Alkaloids

Alkaloids are a large group of nitrogen-containing secondary metabolites with medicinal effects and are found in plants, microorganisms, and animals. To date, more than 27,000 different alkaloids have been identified, with about 21,000 extracted from plants. Alkaloids have one or more nitrogen atoms as primary, secondary, or tertiary amines, which play a major role in compound activity. The degree of alkalinity of alkaloids depends on their molecular structure and the presence and position of functional groups. The bioavailability of alkaloids depends on the amino acid group, which is the fourth type of amine. Relatively few amino acid precursors are involved in alkaloid biosynthesis; the principal ones are ornithine, lysine, nicotinic acid, tyrosine, tryptophan, anthranilic acid, histidine, and phenylalanine. The ability of alkaloids to form salts and complex with metal ions helps in their isolation and identification. Many have potent physiological effects and are important therapeutic agents, e.g., atropine, morphine, and quinine, and are used to treat a wide range of diseases, including malaria and cancer [57]. 

Alkaloids are found in 15–30% of all flowering plants, and the most widely occurring are caffeine and berberine. Over 10,000 different alkaloids have been isolated from over 300 plant families. The most important alkaloid-bearing plant families are *Liliaceae*, *Amarylidaceae*, *Asteraceae*, *Ranunculaceae*, *Papaveraceae*, *Leguminosae*, *Rutaceae*, *Loganiaceae*, *Apocyanaceae*, *Solanaceae*, and *Rubiaceae*. Alkaloids may be found in roots, rhizomes, leaves, bark, fruit, or seeds. Although higher plants are the major source of alkaloids, they are also found in lower plants, such as horsetails, and in algae, fungi (ergotamine and ergometrine from Ergot), microorganisms, insects, and even the organs of mammals (e.g., muscopyridine from musk deer) [58]. Due to these significant properties, the breeding of plants with enhanced alkaloid production is of great importance, and there are many examples of polyploidy being used to this end. For example, scopolamine production was increased in *Hyoscyamus reticulatus* [59], morphine contents in *Papaver somniferum* were enhanced by up to 50% [50], and increases in scopolamine (2.63-fold) and hyoscyamine (1.41-fold) were achieved in the leaves of a tetraploid relative to diploid plants of *Datura stramonium* [60]. Dehghan et al. [7] showed that the persistent autotetraploid plants of Egyptian henbane (*Hyoscyamus muticus*) produced in the fifth generation after tetraploid induction could provide over 200% more scopolamine than diploid specimens. Autotetraploid chicory plants had more phenolic compounds than the controls, producing 10-fold more chlorogenic acid (Table 1) [20]. 

### 6.2. Terpenes

C_10_H_16_ compounds were first named “terpenes” by Kekulé in 1880 because they were isolated from turpentine (Latin balsamum terebinthinae), which emerges from various species of pine trees after the bark or new wood is cut. Although terpenoids are usually regarded as secondary plant products, some of them are involved in basic biological processes such as electron transport in mitochondria and plastids. Terpenes are the most numerous and structurally diverse plant secondary metabolites. More than 55,000 terpenes have been isolated, with the number practically doubling every decade. The diverse functional roles of many terpenoids have been characterized. The smells, tastes, or pharmacological activities of eucalyptus, conifer wood, balm trees, cinnamon, cloves, citrus fruits, coriander, ginger, lavender, lemongrass, lilies, carnation, caraway, peppermint species, roses, rosemary, sage, thyme, violets, and many other plants or their individual parts (stems, leaves, blossoms, roots, rhizomes, fruits, seed) are chiefly attributed to the presence of terpenes. Terpenes have also proved useful as hormones (gibberellins), photosynthetic pigments (phytol, carotenoids), electron carriers (ubiquinone, plastoquinone), and mediators of polysaccharide assembly, as well as in communication and defense mechanisms. Most plants rich in terpenes belong to the families *Apiaceae*, *Asteraceae*, *Cupressaceae*, *Lamiaceae*, *Lauraceae*, *Myrtaceae, Pinaceae, Piperaceae, Rutaceae, Santalaceae,* and *Zingiberaceae* [61].

There are many examples of using polyploidy to enhance terpene production in plants (Table 1). A 56% increase in artemisinin content in tetraploid *Artemisia annua* [53], alleviation of oxidative stress in *Dendranthema nankingense* [62], higher triterpenoid levels in *Centella asiatica* [63], and enhanced tanshinone levels in tetraploid *Salvia miltiorrhiza* [26] have been reported. The essential oil of *Acorus calamus* contained more than 95% anti-cancer beta-asarone in tetraploid plants, whereas it is generally absent in diploid plants [64]. Lin et al. [53] reported that the average level of artemisinin in tetraploid *Artemisia annua* was 39–56% higher than in diploids. The essential oil content increased by 27.5% in the organs of tetraploid plants of *Dracocephalum moldavica* L. [65] and up to 32% in tetraploid *Tanacetum parthenium* L. [66]. The amount of essential oil in autotetraploid plants increased by 30% in *Mentha arvensis* L. and 35–85% in *Carum carvi* L. compared to diploid plants [67,68]. Gao et al. [69] reported up to 78% more total tanshinones and cryptotanshinone in tetraploid lines of *Salvia miltiorrhiza*. The ratio of methyl chavicol, which is the main terpene metabolite of *Agastache foeniculum*, increased in tetraploid plants (81.02%) compared to diploids (78.75%) [70]. Tetraploid plants of *Thymus persicus* showed significant increases in betulinic acid (69.73%), oleanolic acid (42.76%), and ursolic acid (140.67%) compared to diploid plants (Table 1) [71].

### 6.3. Phenols

Phenolic compounds include a large number of molecules with diverse roles in plant growth and development and in defense against insects, fungi, bacteria, and viruses. They include signal molecules, colorants, and flavors. Most phenolic compounds are found in the form of esters, glycosides, or amides, and they are rarely found in their free form [72]. 

Flavonoids are a large group of polyphenols consisting of more than 4000 structurally different compounds, each of which is naturally free or glycosylated [73]. Flavonoids are C15 compounds with a C6–C3–C6 structure. In all of them, two benzene rings are connected to a C3 group, defining how these metabolites are ordered. Flavonoids can be divided into six distinct subclasses: anthocyanins, flavanols, flavones, isoflavones, flavonols, and flavanones [74]. 

Plant lignans, specifically phytoestrogens (pinoresinol, lariciresinol, secoisolariciresinol, syringaresinol, or sesamin), are formed from phenylalanine and can be metabolized to mammalian lignans such as enerodiol and enterolactone by intestinal bacteria [75]. 

Artificial ploidy manipulation has successfully increased phenol contents in plants (Table 1). For example, studies have reported increases in methoxylated flavones in *Dracocephalum kotschyi* [76], valuable compounds including apigenin in *Chamomilla recutita* (L.) [77], and phenylpropanoids in *Solanum commersonii* [78]. Ravandi et al. [42] reported a significantly enhanced level of the total phenolic and chlorogenic acid content in the leaves of tetraploid *Cichorium intybus.* In *Scutellaria baicalensis*, baicalin levels in a tetraploid line were increased by 4.6% [69]. Increasing the ploidy level from diploid to tetraploid resulted in a 1.39- and 1.23-fold increase in podophyllotoxins in the leaves and stems of *Linum album*, respectively [52,54]. In other research, Abdoli et al. [79] reported a 71 and 45% increase in cichoric acid and chlorogenic acid, respectively, in the leaves of tetraploid *Echinacea purpurea* L. Similar results were also reported for artemisinin [53] and phenylpropanoids [80].

## 7. Limitations

Recent studies have provided interesting insights into the regulatory and genomic consequences of polyploidy. However, when polyploidy is induced in plant genomes, it can negatively affect secondary metabolite production [83]. For example, increasing the genome size in relation to the cell volume can alter the balance between chromosomes and nuclear components, resulting in genomic instability [84]. It may also interfere with the correct development of mitosis and meiosis, modifying the chromosome composition of the resulting cells. Furthermore, although an increase in the gene copy number has been demonstrated to positively affect its expression, some regulating factors are not affected in the same way. Thus, gene redundancy can lead to a variety of random outcomes, including a loss of expression [85,86]. In contrast, there is evidence to support redundancy as an opportunity for gene mutation and divergence into new functions [87,88]. 

Variations in molecular regulatory mechanisms involving genomic modifications, such as histone modifications, DNA methylation, miRNA targeting, alternative splicing, and RNA-binding proteins [89], represent a major challenge for the prediction of polyploidy effects on gene expression. These alterations have been investigated in allopolyploids, but they are still understudied in autopolyploids. For example, after genome duplication in the autotetraploid *Chrysanthemum lavandulifolium*, the level of DNA methylation increased [90], but the potential effects of this change on gene expression still require further research. m6A RNA methylation was observed to stabilize mRNA transcripts related to the stress response and has been associated with the regulation of development, photosynthetic processes, and basic metabolism in *A. thaliana* [82,91,92], but this mechanism is still largely unexplored. More research is therefore required to understand the different epigenetic regulatory consequences of polyploidy involving plant secondary production rates.

The use of colchicine for polyploidization in horticultural plants has been practiced for more than 100 years and has recently increased. The limited information available about polyploidy effects and the large amount of data that can be generated from the study of newly generated polyploidies should encourage the creation of specific open-access databases (FAO/IAEA mutant database) [39] for polyploid organisms. Sharing this information could improve our current understanding of the specific effects of polyploidy in plant organisms.

## 8. Conclusions and Perspectives

Polyploidy is a common phenomenon in plants. Ploidy manipulation can also be carried out in vitro with the help of various chemicals, each with its own advantages and disadvantages. Polyploidy is associated with a wide variety of structural, developmental, physiological, and biochemical changes in plants, thereby creating new opportunities for breeders to select optimal plants for a range of purposes, including medicinal and ornamental functions and enhanced resistance. By affecting the enzymatic activity of biosynthetic pathways, the ploidy level alters the quantitative and qualitative patterns of secondary metabolite production. Increasing plant biomass and secondary metabolite yields is an ongoing challenge for researchers. Although the polyploid bridge is a traditional and well-known breeding method, it has been improved by new molecular approaches and biotechnological techniques. The induction of polyploidy in in vitro cell cultures, which involves genome duplication, represents an interesting aspect of tissue culture and plant regeneration and may be applied to increase the production rate of specialized plant metabolites. The use of biotechnological methods, such as somatic hybridization through the mixing of protoplasts, can increase the yield of certain metabolites or lead to the production of new compounds. Polyploidy can also be used as an effective strategy to help to improve the production of organic compounds in vitro, such as hairy roots (especially in cases where the target compounds are produced in the root) or cell suspension cultures, although possible negative effects need to be considered when novel polyploidy organisms are being generated for these purposes.

## Figures and Tables

**Figure 1 biomolecules-11-00899-f001:**
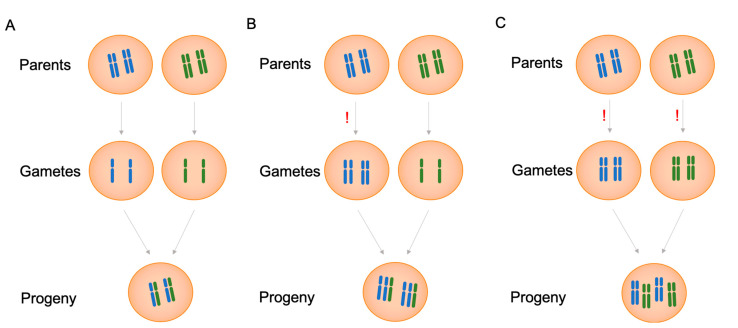
Possible types of gametes after meiosis and offspring formation in autopolyploids. (**A**) Normal meiosis: diploid progeny (fertile). (**B**) Abnormal meiosis autotriploid progeny (sterile) and (**C**) Abnormal meiosis autotetraploid progeny (fertile). Symbol !: nondisjunction in meiosis.

**Figure 2 biomolecules-11-00899-f002:**
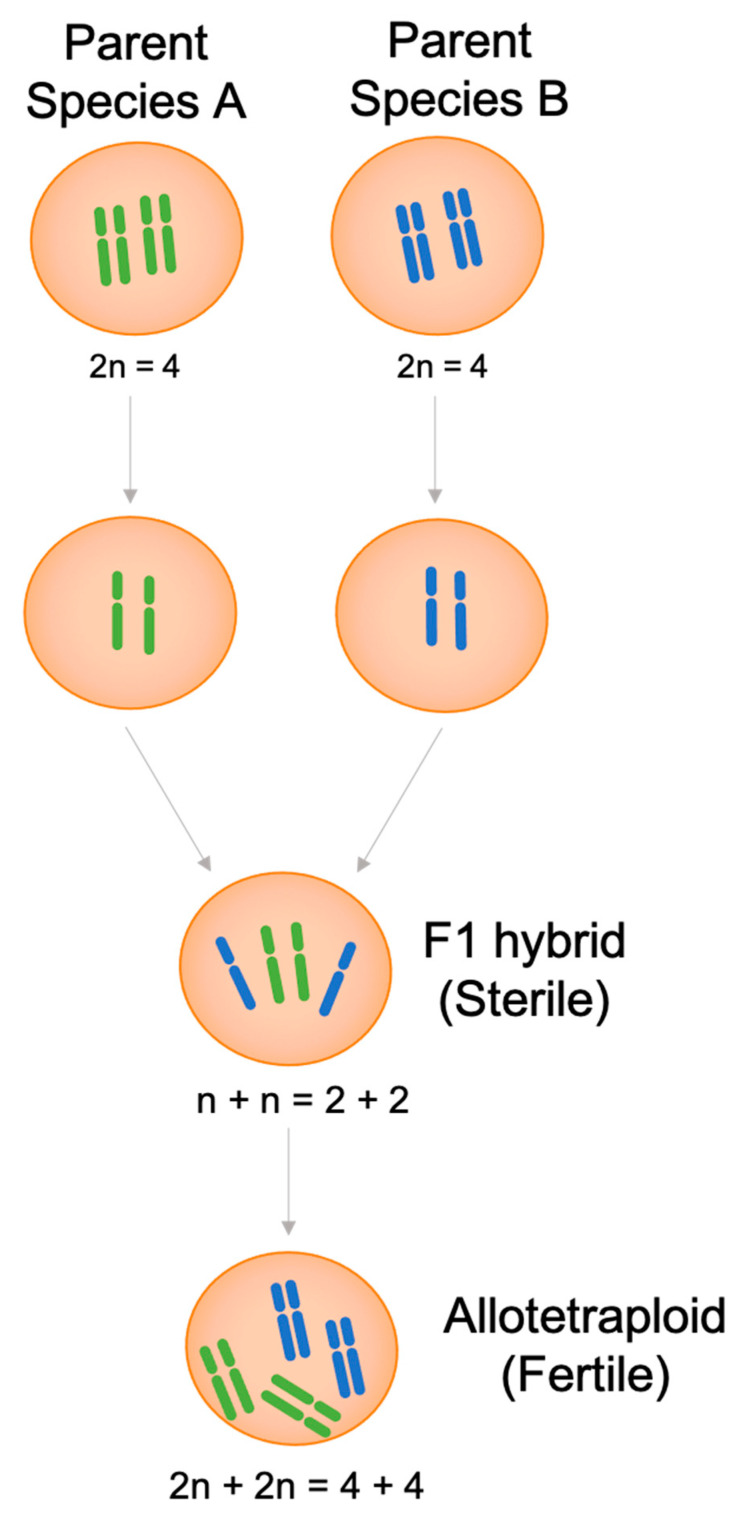
Type of gametes and duplication event during the formation of an allopolyploid.

**Figure 3 biomolecules-11-00899-f003:**
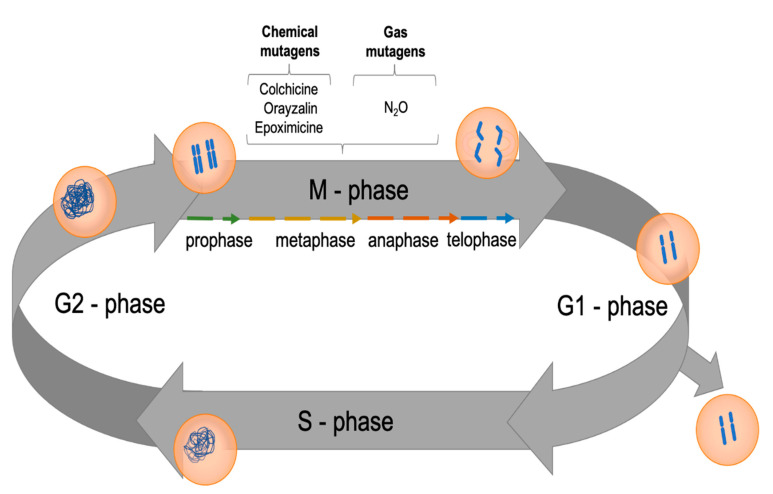
Action of anti-mitotic agents in the plant cell cycle for inducing polyploidy.

**Table 1 biomolecules-11-00899-t001:** Effect of ploidy level change on secondary metabolite production.

Plant Species	Ploidy Level	The Effect of Induced Polyploidy in Diploid Plants	Reference
*Acorus calamus*	*4x*	More than 95% increase in anti-cancer agent beta-asarone	[64]
*Agastache foeniculum*	*4x*	Increase in methyl chavicol metabolite content	[70]
*Artemisia annua*	*4x*	Artemisinin content increased up to 1.5-fold	[53]
*Artemisia annua* L.	*4x*	Artemisinin content increased by 39–56%	
*Papaver bracteatum*	*4x*	5.86- and 30.55-times higher thebaine and noscapine	[81]
*Centella asiatica*	*4x*	Enhanced triterpenoid production	[63]
*Chamomilla recutita*	*4x*	Increased apigenin content	[77]
*Cichorium intybus*	*4x*	Increase in total phenolic compounds and chlorogenic acid	[42]
*Datura stramonium*	*4x*	52–152% increase in alkaloid production	[60]
*Dioscorea zingiberensis*	*4x*	Higher concentration of total alkaloids	[82]
*Dracocephalum kotschyi*	*4x*	Increase in methoxylated flavones	[76]
*Dracocephalum moldavica* L.	*4x*	Higher concentration of essential oil	[65]
*Echinacea purpurea*	*4x*	71 and 45% increase in chlorogenic acid and cichoric acid content	[79]
*Hyoscyamus muticus*	*4x*	200% increase in tropane alkaloids in the fifth generation after tetraploidy	[7]
*Hyoscyamus reticulatus* L.	*4x*	8.8% increase in scopolamine in the plant itself	[59]
*Linum album*	*4x*	1.39- and 1.23-fold increase in podophyllotoxin production	[54]
*Papaver sominferum.*	*4x*	Up to 50% increase in morphine content	[52]
*Scutellaria baicalensis*	*4x*	4.6% increase in baicalin production	[69]
*Solanum commersonii*	*4x*	Higher concentration of phenylpropanoids	[78]
*Tanacetum parthenium* L.	*4x*	Up to 32% increase in essential oil	[66]
*Thymus persicus*	*4x*	Increase in betulinic acid (69.73%), oleanolic acid (42.76%), and ursolic acid (140.6%)	[71]

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
