# Peer review of "Effect of Polyploidy Induction on Natural Metabolite Production in Medicinal Plants"

_biomolecules, 2021, doi:10.3390/biom11060899_

Round 1

Reviewer 1 Report

Review MS titled "Effect of polyploidy induction on natural metabolite production in medicinal plants" by Madani et al. is nice summarized knowledge on the advantages and disadvantages of polyploidy in specialized metabolites production.

Minor revision:

Page 8: I would sugest to change "increase and decrease of genes" to increase the expression (or overexpression) and decrease expression (or downexpression) of genes. It is strange to say increase and decrease of genes. 

Page 9: Plants rich in terpenes mostly belong to the families Apiaceae, Asteraceae, Cupressaceae, Lamiaceae, Lauraceae, Myrtaceae, Pinaceae, Piperaceae, Rutaceae, Santalaceae, and Zingiberaceae [59] - Plant families are not going in italic

If it is possible, authors may add some plant figure (diploid vs polyploid) to demonstrate the effect on morphology etc.

I suggest publishing this review MS after minor revision.

Reviewer 2 Report

The manuscript by Madani et al. attemps to provide a review on the effects of polyploid induction on metabolite production in medicinal plants.

From my view, the article is timely and the topic is of potential interest to colleagues and people working with medicinal plants and/or polyploidy.

As stated in the title, I was expecting to see a detailed survey focused on medicinal plants. Instead, the authors present a broader analysis and include aspects on the causes and consequences of polyploidy in plants. Even when this is not incorrect, most of such information has been repeated in many review articles (e.g., information related to the use of anti-mitotic agents in past decades) and is presented elsewhere with more detail, and is therefore of relative relevance here.

In addition, many sentences lack clarity and place statements without providing references.  
Having a review focusing on the progress and advances in the understanding of polyploid effects on metabolite production in medicinal plants could be novel to the public. However, this is not yet provided in the present version of the manuscript.

I am not a native speaker, but having the text read by a professional editing and proofreading service will improve substantially the manuscript. As it is now, many sentences are hard to read and it is not possible to follow the rational behind. Below I point out a few examples.

More specific comments and some editing can be found in the attached manuscript file.

General comments:

Section 2 could be substantially reduced. 

On page 4, the meaning of "mixed chromosome sequences" is unclear.
Even if chromosomes may undergo some exchange of sequences related to mobile elements, it is not clearly stated and it does not seem relevant here.

Similarly, the use of phrases like 

"as these [chromosomes] are not homologous and cannot be synaptic"

"the first-generation hybrid will be germinal", just to mention another two, is not completely correct.

"allopolyploids had been successfully used to introduce beneficial alien genes"....how exactly? if by "alien genes" is mean a gene from another species, then that seems to be part of the definition of an allopolyploid, i.e. a polyploid derived from two or more species.

with "overcome cross incompatibility by ensuring compatible ploidy
status in the prospective parents,"
the authors mean overcoming interploid (or heteroploid) barriers?

Sentences like this

"To differentiate between the sources of the genomes in an alloploid,
each genome is designated by a different letter. For example, the origin of the cultivated mustards (Brassica spp) is explained by the triangle of Nagaharu U, with each species represented by a distinct letter [34]."

(on page 4), or this 

"During the metaphase, a canal is created from the alpha and beta microtubules for the transfer of chromosomes during the anaphase [41]."

(on page 6) are hard to follow and misleading.

The reading of the first one suggest genomes are differentiated by giving them a letter. What the authors might want to say is that genomes are differentiated by homology (either at chromosomal or sequence levels), and letters are assigned in consequence (but only for practical purposes);

In the second one, an apparently erroneous view of the existence of a channel involved in chromosome transfer is given. Even when microtubuli are tubular structures, they only attach to proteins in the kinetochore and pull the chromosomes (sister chromatids) to each pole during mitosis. There is no "canal" involved. 

Similarly, the use of names like "the growth retrieval method" (on page 6) is somehow vague. I have not found any information on the internet about it.

Author Response

"Please see the attachment

Round 2

Reviewer 2 Report

The authors have made substantial changes in the new version.

However, I still think you should have the manuscript read by a copy/editing professional service (not the same as a native speaker). As example, you can check a few copy/editing points added in section 5 which are needed to make the manuscript sound.

In addition, still there are several terms used wrongly or ambiguously (and not only those I mentioned in the first round which were corrected or eliminated) which weaken the readability of the piece. I cannot list them all here. For example, in the abstract, the use of 'polyploidy methods' is incorrect. 'Polyploidy' is a condition, not a method.

or at the end of section 7, the expression 'the global mutant production status of ecologically and commercially significant plant species'. is vague.

Other phrases like 'Colchicine, oryzalin, and trifluralin have been used as antihyperglycemic agents in intraperitoneal polyploidization studies'. simply do not fit in a review on plants without providing a clear explanation and references supporting it.

What would be the difference between 'future work' and 'perspectives' in sections 7 and 8?

Round 3

Reviewer 2 Report

I appreciate the effort made by the authors. The new version of the review has been substantially improved and the streamlined.

I found some typos.

"Materia Medica" not "Material Medica"

in Figure 2 it should be "n + n = 2 + 2", not "n + n - 2 + 2"

in page 9: "During in vitro induction..."

in page 15: "...production of organic compounds in in vitro cultures such as..." is redundant, better "production of organic compounds in vitro such as..."

A comment for Figure 1. This figure deals with autopolyploidy. In B.1, having 6 chromosomes grouped in three is correct (for an organisms with two chromosomes as depicted here). However, that also implies that you should have chromosomes grouped in four in B.2 (which is not yet the case). This is not critical, of course. But you could improve the figure by drawing two different chromosomes for the organism, for example, using different colours or structure, i.e., one pair could be a metacentric (as now) and the other could be an acrocentric. Then, the groups could be easily identified. 

I should mention (once more) that in different places along the text, references are still needed.

For example, in page 9 you wrote "Studies on Arabidopsis plants have shown that some of the genes that were active in a the diploid state were shut down in the autotetraploid state, but the cross between two tetraploid Arabidopsis species led to the production of allotetraploid plants in which these genes reactivated."

Since this is not something derived from the review, clearly, I would appreciate -as a reader- to be able to identify from which work(s) such statement is coming from. That is the reason why I would read a review in the first place (and would make your review more sound).

The same in page 10 and other pages. "Recent experiments demonstrate that the various ploidy levels levels and the formation of polyploids in a plant (autoploids or alloploids) and even the previous generations of a polyploid plant play a role in the level and stability of gene expression. There is considerable discussion about the factors that affect the activity of genes and their expression in polyploids, including the activation of dormant transpositions in synthetic polyploids, which can causes the extinction of gene."

These are two strong sentences for which not a single reference is added nor it is possible to connect them to a reference. Without a reference supporting your statements, they become anecdotic.

Author Response

Response to Review comments.

Thank you very much for all your comments and suggestions to improve the quality of our manuscript. We have changed the previous version according to all your suggestions.

Comment 1:"Materia Medica" not "Material Medica"

Response 1: Sorry for the mistake, the change has been done.

Comment 2: in Figure 2 it should be "n + n = 2 + 2", not "n + n - 2 + 2"

Response 2: Sorry for the mistake, the figure has been modified according to your suggestion.

Comment 3: in page 9: "During in vitro induction..."

Response 3: The change has been done.

Comment 4: in page 15: "...production of organic compounds in in vitro cultures such as..." is redundant, better "production of organic compounds in vitro such as..."

Response 4: The sentence has been modified according to your suggestion.

Comment 5: A comment for Figure 1. This figure deals with autopolyploidy. In B.1, having 6 chromosomes grouped in three is correct (for an organisms with two chromosomes as depicted here). However, that also implies that you should have chromosomes grouped in four in B.2 (which is not yet the case). This is not critical, of course. But you could improve the figure by drawing two different chromosomes for the organism, for example, using different colours or structure, i.e., one pair could be a metacentric (as now) and the other could be an acrocentric. Then, the groups could be easily identified.

Response 5: Thank you for the suggestion. We are in complete agreement, and the Figure has been modified to clarify it.

Comment 6: I should mention (once more) that in different places along the text, references are still needed.

Response 6: Following your suggestion, new references have been added and the reference numbering has been changed accordingly.

Comment 7: For example, in page 9 you wrote "Studies on Arabidopsis plants have shown that some of the genes that were active in a the diploid state were shut down in the autotetraploid state, but the cross between two tetraploid Arabidopsis species led to the production of allotetraploid plants in which these genes reactivated NG."

Since this is not something derived from the review, clearly, I would appreciate -as a reader- to be able to identify from which work(s) such statement is coming from. That is the reason why I would read a review in the first place (and would make your review more sound).

Response 7: A new reference has been added after this paragraph.

Comment 8: The same in page 10 and other pages. "Recent experiments demonstrate that the various ploidy levels and the formation of polyploids in a plant (autoploids or alloploids) and even the previous generations of a polyploid plant play a role in the level and stability of gene expression. There is considerable discussion about the factors that affect the activity of genes and their expression in polyploids, including the activation of dormant transpositions in synthetic polyploids, which can causes the extinction of gene.

These are two strong sentences for which not a single reference is added nor it is possible to connect them to a reference. Without a reference supporting your statements, they become anecdotic.

Comment 7: Two references have been added after each sentence.
